# Molecular Phylogeny, DNA Barcoding, and *ITS2* Secondary Structure Predictions in the Medicinally Important *Eryngium* Genotypes of East Coast Region of India

**DOI:** 10.3390/genes13091678

**Published:** 2022-09-19

**Authors:** Gobinda Chandra Acharya, Sansuta Mohanty, Madhumita Dasgupta, Supriya Sahu, Satyapriya Singh, Ayyagari V. V. Koundinya, Meenu Kumari, Ponnam Naresh, Manas Ranjan Sahoo

**Affiliations:** 1Central Horticultural Experiment Station, ICAR–Indian Institute of Horticultural Research, Bhubaneswar 751019, Odisha, India; 2ICAR Research Complex for Northeastern Hill Region, Manipur Centre, Imphal 795004, Manipur, India; 3All India Institute of Medical Sciences, Bhubaneswar 751019, Odisha, India; 4ICAR Research Complex for Eastern Region, Research Centre, Ranchi 834010, Jharkhand, India; 5ICAR–Indian Institute of Horticultural Research, Bengaluru 560089, Karnataka, India

**Keywords:** DNA barcoding, *ITS2* secondary structure, medicinal herbs, species discrimination, spiny coriander

## Abstract

Commercial interest in the culinary herb, *Eryngium foetidum* L., has increased worldwide due to its typical pungency, similar to coriander or cilantro, with immense pharmaceutical components. The molecular delimitation and taxonomic classification of this lesser-known medicinal plant are restricted to conventional phenotyping and DNA-based marker evaluation, which hinders accurate identification, genetic conservation, and safe utilization. This study focused on species discrimination using DNA sequencing with chloroplast–plastid genes (*matK*, *Kim matK*, and *rbcL*) and the nuclear *ITS2* gene in two *Eryngium* genotypes collected from the east coast region of India. The results revealed that *matK* discriminated between two genotypes, however, *Kim matK*, *rbcL*, and *ITS2* identified these genotypes as *E. foetidum*. The ribosomal nuclear *ITS2* region exhibited significant inter- and intra-specific divergence, depicted in the DNA barcodes and the secondary structures derived based on the minimum free energy. Although the efficiency of *matK* genes is better in species discrimination, *ITS2* demonstrated polyphyletic phylogeny, and could be used as a reliable marker for genetic divergence studies understanding the mechanisms of RNA molecules. The results of this study provide insights into the scientific basis of species identification, genetic conservation, and safe utilization of this important medicinal plant species.

## 1. Introduction

Spiny coriander (*Eryngium* spp.), one of the largest taxonomically complex genera in the order, *Apiales*, and family, *Apiaceae*, is a perennial herb used as food and medicine in the world’s tropics. The taxonomical evolution of the genus, *Eryngium*, is coupled with rapid radiation, hybridization, and long–distance dispersal. *Eryngium* comprises about 250 species with a cosmopolitan distribution worldwide, and South America is believed to be the center of diversity [1]. *Eryngium foetidum* L. is known as spiny or false coriander or culantro, and is used as a culinary herb and traditional medicine in Latin America and South-East Asia, having a similar taste and odor to coriander or cilantro. *E. foetidum* is widely cultivated in South-East Asia, tropical Africa, the Pacific Islands, and the southern tropics of Europe [2]. This perennial herb, *E. foetidum*, hailed from tropical America and the Caribbean Islands, and was further dispersed to South-East Asian countries in the 18th century [3]. *E. foetidum* is cultivated widely in the eastern and northeastern states, including the east coast and the Andaman and Nicobar Islands of India [4]. Despite its wide adaptability, the domestication and evolution of *Eryngium* are still debatable.

The leaves and pseudo stems of *E. foetidum* have multiple pharmaceutical uses, such as anti-diabetic [5], anti-inflammatory [6], anti-cancerous [4], and anti-clastogenic [7] properties. The phytochemical constituent eryngial in *Eryngium* leaves, the reason for the typical pungency, is also used for treating skin disease, arthritis, parasites, and other physiological diseases [4,8]. A mixture of the shoot and root portion is used as a folk medicine worldwide, such as an appetizer, antitussive, treating diarrhea and gastrointestinal disorders, hypoglycemic, and anti-venom agent [8]. Apart from treating several ailments, *Eryngium* is a potent spice for food garnishing and a flavoring agent in the coastal region of India. A recent study depicted that the component, 10-undecenal, is the reason for a strong aroma similar to the seasonal coriander in the *Eryngium* spp. collected from the east coast of India [9]. This perennial lesser-known herb possesses high chemo diversity importance for the pharmaceutical and aroma industries.

The use of traditional herbal medicine is increasing day by day. The correct nomenclature and accurate species identification need the utmost attention for the safe use and commercialization of medicinally important plants. *Eryngium* is a hemicryptophyte perennial herb with spiny leaf margins, light blue inflorescence, and a tap root system [9]. Reports on molecular characterization for species discrimination in *Eryngium* are scanty. The subgenera and species of *Eryngium* have been categorized based on their morphological features, such as leaf margin, color, odor, and chemical constituents [9,10]. A few reports are available on the infrageneric relationship using chloroplast DNA (*trnQ–trnK*) and nuclear internal transcribed spacer (*ITS*) sequencing [11]. Considering morphological variability, species discrimination in closely related genera is quite difficult. Molecular characterization using random DNA markers showed a compressive genetic variability among the genotypes. However, DNA barcoding is an advanced tool to investigate variations at the most conserved region to specify accurate species delimitation [12]. 

DNA barcoding is a robust emerging technique that confirms species boundaries from a small plant tissue using a short DNA section from a specific gene or genes [13], recommended by the Consortium for the Barcode of Life (CBOL) plant group [14]. The gene region includes a chloroplast–plastid region (*matK*, *rbcL*, *ycf*, *psbA–trnH*, etc.) and a nuclear *ITS* region with less intraspecific barcode gaps [15]. Maturase K (*matK* (XF/5R)) and *Kim matK* (3F_KIM/1R_KIM), the plant plastidial gene in the chloroplast region that splices group II introns in the conserved domain X of the reverse transcriptase domain can be used as a universal DNA barcode primer for plants [16]. The chloroplast DNA sequence, ribulose–bisphosphate carboxylase (*rbcL*), is one of the universal barcode genes used in species discrimination studies due to high amplification and low mutation rate [17]. *Ycf1* is the second largest gene in the plastid genome, encodes about 1800 amino acids, and is a reliable candidate for DNA barcoding [18]. The *psbA–trnH* gene is a non-coding fragment between *psbA* and *trnH* in the chloroplast genome, which delimitates faster evolution than *matK* and slower than *ITS*. The *ITS* region is a DNA spacer, localized between the small and large subunits of ribosomal RNA (rRNA) in the chromosomal or corresponding polycistronic transcript region, most commonly used for species discrimination studies [19]. The CBOL–Plant working group, suggested a combination of plastid (*matk*/*Kim matK*/*rbcL*) and nuclear region (ITS) as the efficient barcode tool to investigate plant species discrimination [20].

RNA secondary structure predictions at the conserved ITS rRNA region is a key ribosomal structure that predicts the function of rRNAs and tRNAs [21]. Computationally-predicted RNA structures represent the native RNA folding status of an organism, shedding light on novel RNA regulatory mechanisms [22]. RNA secondary structure prediction is an advanced tool for species discrimination, as it restricts sequencing error and eliminates pseudogene footprints [23].

The present study aimed to discriminate two important *Eryngium* genotypes grown in two vulnerable coastal agroclimatic regions of India using *matK*, *Kim matK*, *rbcL*, and *ITS* DNA barcode genes. We have also evaluated the potential of different barcode primers at chloroplast–plastid and nuclear regions discriminating the *Eryngium* spp. The phylogenetic relationship among these two *Eryngium* genotypes was established concerning the barcode regions. To elucidate the variation among the two contrast genotypes, *ITS2* secondary structure prediction was performed to validate the signature barcodes. The result of this study provides insights into the scientific scope for species identification in this lesser-known crop, using DNA barcoding as a robust tool for understanding genetic relationship studies for future crop improvement strategies.

## 2. Materials and Methods

### 2.1. Plant Materials

The two most popularly grown *Eryngium* genotypes were collected from the east and southeastern coastal plain zone of Odisha, Delanga, India (voucher specimen: Eryngium_CHB, and biological reference number: IC 0629514), and from the ICAR–Central Island Agricultural Research Institute (CIARI), the Island region of Car Nicobar, Port Blair, India (voucher number: Eryngium_AND), and maintained at the Central Horticultural Experiment Station (CHES), ICAR–Indian Institute of Horticultural Research (ICAR–IIHR), Bhubaneswar, India, and were used as the source materials for the present study. The study area is located at a latitude of 20°15′ N, a longitude of 85°52′ E, and an altitude of 35 m above mean sea level. The geographical location of the center of the collection of the two *Eryngium* samples is presented in Figure 1.

### 2.2. Plant Sample Preparation

The first fully opened juvenile leaf samples were collected from the two *Eryngium* genotypes grown under a naturally ventilated poly house. The herbarium for the voucher specimen numbers, Eryngium_CHB and Eryngium_AND, was prepared following the standard procedures and submitted to the repository of ICAR–IIHR–CHES. A set of fresh leaf samples from the respective genotypes were used for DNA barcoding and RNA secondary structure prediction studies.

### 2.3. Total Genomic DNA Isolation

Total genomic DNA (gDNA) was isolated from the fresh leaf tissues of the two *Eryngium* genotypes using a plant gDNA extraction kit (GSure^®^ Plant Mini Kit with WLN Buffer, GCC Biotech Pvt Ltd., Kolkata, India) following the manufacturer’s instructions. The gDNAs were quantified using a nanodrop (Eppendorf, Hamburg, Germany) and checked on 0.8% agarose gel electrophoresis (Tarson, Kolkata, India). The gDNA concentration was adjusted to 50 ng µL^−1^, and was used for PCR amplification with the DNA barcode primers, *matK*, *Kim matK*, *rbcL*, and *ITS*, as recommended by the Consortium of Barcode of Life–Plant Group [14]. The details of the barcode primers are presented in Table 1.

### 2.4. PCR Amplification

The DNA barcode primers, *matK*, *Kim matK*, *rbcL*, and *ITS*, were synthesized at M/S Bioserve Biotechnologies India Pvt. Ltd., Hyderabad, India. PCR amplification for each primer was performed in a volume of 25 µL, containing 50 ng of gDNA (1 µL) as a template, 12.5 µL 2 × PCR master mix (GCC Biotech Pvt Ltd., Kolkata, India), primers (10 pM, 1 µL each of forward and reverse primers), and 9.5 µL Milli–Q water following Ayaz et al. (2020) [24] with partial modification. The PCR amplifications were carried out in a thermal cycler (Eppendorf, Hamburg, Germany). The PCR conditions follow a denaturation of 5 min at 95 °C, 40 cycles of 1 min at 95 °C, 1 min at 55 °C of annealing, 1 min at 72 °C, and a final extension of 10 min at 72 °C. Purification of the PCR products was performed using a PCR purification kit (GCC Biotech Pvt. Ltd., Kolkata, India) by following the manufacturer’s instructions. The purified PCR products were visualized in 1.5% agarose TAE gels in an E–Box gel documentation system (Vilber, Eberhardzell, Germany). Good-quality PCR products (50 ng) were considered for further sequencing.

### 2.5. DNA Sequencing

The purified PCR products were sequenced at M/S Bioserve Biotechnologies India Pvt. Ltd., Hyderabad, India, using Sanger sequencing (ABI Genetic Analyzer 3730, 48 capillaries, 50 cm, Thermo Fisher, Waltham, MA, USA), and viewed in FinchTV v1.4.0 (Geospiza, Denver, CO, USA).

### 2.6. Bioinformatics Analysis

The forward and the reverse sequences obtained from all the PCR products amplified with *matK*, *Kim matK*, *rbcL*, and *ITS* primers were trimmed using SnapGene v 5.3 (https://www.snapgene.com; accessed on 4 September 2022). The contig of the forward and the reverse sequences was submitted in the Basic Local Alignment Search Tool (BLAST) of the National Centre for Biotechnology Information (NCBI) for homology search. The barcode gaps were manually edited in pairwise alignment view using BLAST, and species identification was performed following nucleotide blast with the maximum similarity score and lowest E value. The gene bank accession number was obtained using the barcode sequences in the bankIt submission portal of NCBI (https://submit.ncbi.nlm.nih.gov/subs/genbank; accessed on 21 July 2022). Multiple sequence alignment was performed using ClustalW v10.1.8 (https://www.genome.jp/tools-bin/clustalw; accessed on 4 September 2022) [25] with all obtained sequences in “muscle algorithm” using the neighbor-joining cluster method in MEGA11 software (Molecular Evolutionary Genetic Analysis; https://www.megasoftware.net; accessed on 4 September 2022). The phylogenetic analysis was performed following the neighbor-joining tree and minimum evolution method with the 1000 “*Bootstrap phylogeny*” test method and “*kimura–2–parameter*” substitution model (*d–transitions*) in MEGA software, considering the transitional and transversional nucleotide substitution [26].

### 2.7. DNA Barcoding and ITS2 Secondary Structure Predictions

DNA barcodes of the two *Eryngium* genotypes were generated using the Bio-Rad DNA barcode generator (http://biorad-ads.com/DNABarcodeWeb; accessed on 4 September 2022), considering the aligned DNA nucleotide sequences of *ITS2* primers. Similarly, the *RNA* secondary structure predictions were performed using the nucleotide sequences from *ITS2* primers using the rRNA database of *RNAfold* WebServer v2.4.18 (http://rna.tbi.univie.ac.at/cgi-bin/RNAWebSuite/RNAfold.cgi; accessed on 4 September 2022) [27].

## 3. Results

### 3.1. Amplification, Sequencing, Multiple Sequence Alignment, and Species Identification

The DNA barcode primers, *matK*, *Kim matK*, *rbcL*, and *ITS*, produced amplicons of 1500 bp, 1500–1580 bp, 600–800 bp, and 400–800 bp, respectively. The success rate of PCR amplification was higher (>90%) in *matK*, *Kim matK*, and *rbcL*, whereas *ITS2* showed lower reaction efficiency (80%). The standardization of template DNA concentration and specific primer at different ITS regions are required to validate the amplification success rate. Analyzing the sequences, *matK* exhibited the largest sequence length (534–823 bp), followed by *Kim matK* (763–780 bp), *rbcL* (476–493 bp), and *ITS* (300–357 bp). All the sequences were submitted to GenBank, and the accession numbers were obtained. The details of voucher specimens, species, and accession numbers are presented in Table 2. Using the BLASTn tool, both species were identified as *E. foetidum* at different barcode regions of *Kim matK*, *rbcL*, and *ITS2*. However, the *matK* gene successfully discriminates the species among the two genotypes. The specimen, Eryngium_AND, was showing similarities with *E. vaseyi*, whereas Eryngium_CHB was identical with *E. foetidum*, with 99% similarity at the conserved *matK* region. 

Figure 2 depicts the multiple sequence alignments obtained from all sequences at each chloroplast–plastid and nuclear region. Among the tested barcode primers, *matK* resulted in the largest alignments (583 bp), followed by *Kim matK* (780 bp), *rbcL* (493 bp), and *ITS* (357 bp). Alignments obtained from *ITS* showed the highest similarities (100%) among the two genotypes, whereas *matK* discriminated two genotypes as *E. vaseyi* (Eryngium_AND; ON797393) with 99.76% identity and *E. foetidum* (Eryngium_CHB; ON797394) with 99.81% identity (Figure 2).

### 3.2. Phylogenetic Studies

Phylogenetic analysis using a maximum likelihood tree (MLT) in a *K2P* model with bootstrap-1000 of *Eryngium* species showed a high similarity in the BLAST search to the available sequences in the NCBI database (Figure 3). Phylogenetic analysis revealed that Eryngium_AND_*matK* (823 bp) had >99.76% similarity with *E. vaseyi* (OP086070), which belongs to the class of *Magnoliopsida*. Other closest relatives of this species include *E. armatum* (OL689984.1), *E. yuccifolium* (MK520075.1), with 99.88% and 99.60% similarity, respectively, as per the BLAST search results. Similarly, the specimen, Eryngium_CHB_*matK* (534 bp), showed the closest similarity, >99.81% to *E. foetidum*, as its closest phylogenetic relative. The other closest matches are *E. baldwinii* OSBAR (MH551983.1), *E. ebracteatum* BioBot01719 (JQ586532.1), *E. cuneifolium* voucher FLAS: Judd 5564 (KY607232.1), with an average similarity of 99% in all exceeding species. The voucher specimens, Eryngium_CHB_*Kim matK* (780 bp), Eryngium_AND_*Kim matK* (763 bp), Eryngium_AND_*rbcL* (493 bp), Eryngium_CHB_*rbcL* (476 bp), Eryngium_AND_CHB (357 bp), and Eryngium_AND_*ITS* (300 bp), belong to *E. foetidum*, with the closest similarity of 99.74%, 99.87%, 100.00%, 100.00%, 100.00%, and 100.00%, respectively, belong to the class of *Magnoliopsida* and the family of *Apiaceae*. The other closest relatives of Eryngium_CHB_*Kim matK* are *Sanicula canadensis* isolate AD3HK81 maturase K (*matK*) gene (MF350053). Eryngium_AND_*rbcL* showed similarity with *Sanicula graveolens* voucher HU WAP 14409 (KX371924), having a similarity of 98.58%. Eryngium_CHB_*ITS* shows the highest similarity (98.88%) with *E. balansae ITS1*, 5.8S ribosomal RNA gene (EU070608). The phylogenetic analysis indicates that the plant genotypes collected from two locations showed species variation with different percentage indexes (Figure 3).

### 3.3. DNA Barcoding and ITS2 Secondary Structure Predictions

Figure 4 represents the DNA barcodes and *ITS2* secondary structure predictions based on the minimum free energy (MFE). The highest MFE for Eryngium_AND (ON797393) was observed at 145–150 bp (Figure 4a; Table 3), which was recorded highest at 275–280 bp in Eryngium_CHB (ON797394) [Figure 4b]. The partition function and centroids followed a similar pattern as the MFE in both genotypes.

DNA barcodes derived from the *ITS2* sequences showed variations among the two tested genotypes (Figure 4c,d). Eryngium_CHB (ON797394) exhibited comparatively higher barcode length (356 bp) compared to Eryngium_AND (ON797393) [299 bp]. Similarly, *ITS2* secondary structure predictions discriminate both the tested genotypes representing a central ring with different helical orientations (Figure 4e,f). The loop number, position, size, and angle from the centroid are distinguishable in both genotypes. Eryngium_CHB (ON797394) represented a more complex structure than Eryngium_AND (ON797393), with varied loop numbers and angles from the spiral. The unique genetic structure at the conserved nuclear region would also be useful to develop species-specific primers to identify the lesser-known species at a shorter pace. The secondary structure predictions guide comparative sequence analysis and design of species–specific RNA molecules.

## 4. Discussion

Perennial *Eryngium* is an important food and medicinal herb gaining significant industrial and economical values in pharmaceutical industries. The medicinal use of traditional herbal plants needs accurate identification for their safe use in alimenting human diseases [28]. DNA barcoding, at the chloroplast–plastid and nuclear regions or in a combination of both regions, is a newer tool receiving increasing attention over the conventional phenotyping-based taxonomy. Molecular signatures at the conserved barcode region provide insights into the genetic importance and possible scope for crop improvement. 

Our study aimed at species discrimination in the lesser-known *Eryngium* genotypes collected from two different coastal microclimatic zones of India by deploying DNA barcode genes. This study is the first report to identify *Eryngium* in this region using chloroplast–plastid and nuclear regions. A few reports [29,30] suggest the plastid barcode gene significantly creates variation at the species level. However, reports on species discrimination in *Eryngium* at both the plastid and nuclear region is scanty. Our study revealed significant variations at the plastid and nuclear regions with 99–100% similarities. Discrimination efficiency was higher in *matK* than *Kim matK*, *rbcL*, and Nuclear *ITS2*. Of the barcode genes tested, only *matK* could perform species delimitation (Figure 3; Table 2), which could be further used for large-scale species discrimination studies. The results of the study would be helpful for a reference database for accurate species identification in *Eryngium*.

DNA barcoding technology is an advanced method of identifying plant species with conservational and consumer marketable value [31]. The abundant relatives of the plant species fulfil the demand for applied herbal medicines obtained from medical plant genotypes. Likewise, amplified demand for medicinal products may lead to the over-harvesting and extinction of the non-threatened biological species. Therefore, identifying the plant samples is urgently needed for the conservation and prevention of perceived biodiversity loss. Thus, appropriate and prominent measures could provide a reliable genetic method for systematically identifying medicinal plant valuables for species preservation and genetic improvement.

Generating the importance of DNA barcoding onto the well-defined gene from genetic regions in a medicinal plant sample and its appearance of species in a specific manner are being studied in this research. To increase the prominence selection in monitoring, the discrimination based on DNA-based identity is found authentic for any medicinal plant specifically related to drug delivery. Specific universal DNA markers/primers for plants recommends genes from the chloroplast region, such as *Kim matK* maturase K (*matK*) gene; maturase K (*matK*) gene; ribulose 1,5-bisphosphate carboxylase/oxygenase large subunit (*rbcL*) gene marker, belonging to the enzyme (RuBisCo); and internal transcribed spacer (*ITS*) primer are selected as a core marker for the barcoding of DNA for medicinal plants [32]. However, this marker does not show 100% discrimination of the plant species, but *matK* is recommended as the marginal selection of the variation found in our collected *Eryngium* species. Ayaz et al. [24] performed genetic diversity studies in the *Lamiaceae* family using *rps 14* genes following molecular phylogeny. DNA-based markers exhibited significant variation among the species; however, barcode markers recommended by CBOL confirm the discriminations at conserved barcode regions.

The main goal of DNA barcoding is to assemble the reference library at a shorter pace, which includes DNA extraction, amplification, and sequencing. We have investigated the discrimination efficiency of four barcode regions (*matK*, *Kim matK*, *rbcL*, and *ITS*) to observe species discrimination, which varied between 80–90%. *matK* resulted in a higher sequence length to discriminate the voucher specimen into two different species, *E. vaseyi* and *E. foetidum*. We have estimated the amplicon universality and the success rate of amplification of various chloroplast–plastid (<90%) and nuclear genes (80%) in the rare taxa, *Eryngium*. Previous reports showed a low amplification rate of 60% in *matK* for species discrimination in other medicinal plants [12]. Molecular phylogeny analyzes the genetic heredity based on the DNA sequences to understand the evolutionary interrelationship, deriving the genetic diversity among the species. Molecular phylogenetics following the maximum likelihood tree depicts the molecular systematics of taxonomy evolution from different geo-coordinating regions [33]. Figure 3 distinguishes the molecular phylogeny based on the maximum likelihood neighbor-joining tree using the minimum evolution method with the bootstrap-phylogeny-1000 method and d-transitions substitution model. Various genetic divergence studies emphasize the phylogenetic exploration of medicinally important plants of the *Lamiaceae* family [24,28]. Among the conserved barcode genes, *ITS2*-anchored evolution, structure, and function establish a better understanding of the species through sequence, structure, and phylogeny based on the nucleotide sequences and independent DNA evolutionary models [34]. In our study, *ITS2*-based phylogeny distinguished the species into a different clade with 100% similarities (Figure 3).

According to the diversity-based discrimination on a species-specific level, the major explanation is that barcodes are improved candidates for divergence in a reliable conserved DNA sequence [35]. The specific approach in the level of discrimination and universality for medicinal plants is found to be more precise with the ribosomal RNA maturase K (*matK*) marker, which is considered a core DNA barcode spacer which would be exclusive in identifying the plant genotypes at the species level. Moreover, the mitochondrial genome arrangement of plants is genetically altered quickly, evading the existence of universal markers amplifying the coding genes (CDS region). Researchers [36,37] investigated the improved discrimination efficiency of different plant species using the combination of *rbcL* and *matK*, and *rbcL* and *ITS2* markers for more suitable variation. Supporting the intensified possibility of the *matK* DNA barcode marker, genetic data infested the quality of divergence [38].

The RNA secondary structure determines the understanding between the structural and functional relationship in designing the therapeutics and diagnostics of the target RNA. Minimum free-energy-anchored RNA secondary structure prediction indicated that site-specific mutagenesis occurred due to tertiary protein interaction. The ribosomal RNA structure has the advantage of understanding RNA chemistry compared to the nuclease mapping [39]. In our study, both genotypes represented diverse secondary structures, with distinguishable loop numbers, positions, and elevation from the centroid. The unique nuclear segments can be further used to develop species-specific markers for identifying the genotypes. *ITS2* could not show different variations at the species level, but could predict the differential secondary structure, indicating variations among the RNA molecules in both genotypes. The study would encourage RNA molecule studies for genetic evolution analysis of plant taxa.

In general, molecular approaches appear to be an ideal advancement suited to categorizing indistinguishable plant species. In supplementing the DNA sequencing database to the gene bank, reliably identified species exist for the barcode reference for the chosen plant genotypes for the possible implication of informatics in future medicinal analysis. The DNA database can target a broad range of conservational and ethnobiological traditional potentiality of this underutilized medicinal crop. For the participatory improvement of the crop, the medicinal germplasm is systematically maintained in a sequence database for future quality yield with the genetic variation. The suitable application of the traditional products concerning genetic complexity and identified plant samples can address a good companion integrated with conserving this medicinal plant.

## 5. Conclusions

The DNA barcoding results at the chloroplast–plastid and nuclear region discriminate the two *Eryngium* species. The plastid gene, *matK*, could be a potent barcode gene to discriminate between two genotypes, which may be further used for a large-scale divergence study. However, the ribosomal nuclear *ITS2* region could be a reliable nuclear region to distinguish inter- and intra–specific divergence, DNA barcoding, predictions of the secondary structure, and understanding and re-designing RNA molecules. The results of this study provide insights into the scientific basis of species identification, genetic conservation, and safe utilization of this important medicinal plant species. The DNA barcode tools could be used for species delimitation in medicinally and commercially important plants, and could also be helpful in the detection of adulteration in food and pharmaceutical industries. The results also encourage understanding genetic relationships for future crop improvement strategies for food, nutrition, and therapeutics. 

## Figures and Tables

**Figure 1 genes-13-01678-f001:**
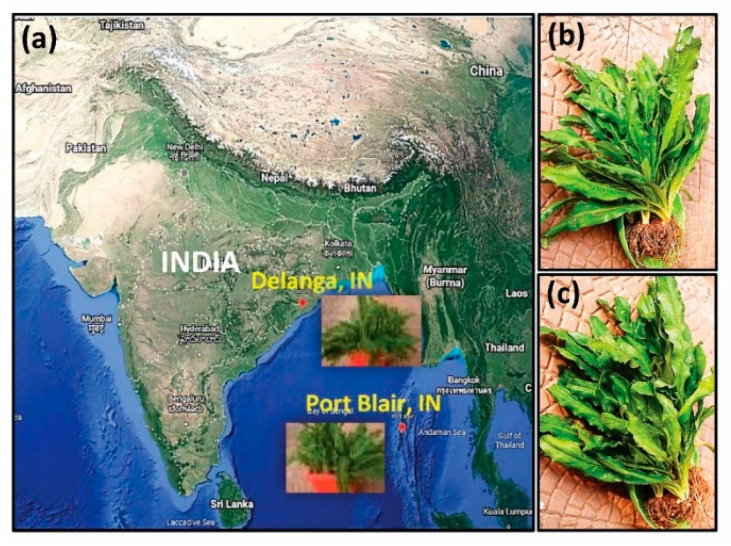
The geographical location (**a**) of the *Eryngium* genotypes collected from Delanga (20°03′ N, 85°76′ E, and 0 masl), Odisha, India (**b**), and Port Blair (11°62′ N, 92°72′ E, and 16 masl), Andaman and Nicobar Island, India (**c**) (source: http://earth.google.com/web, assessed on 4 September 2022).

**Figure 2 genes-13-01678-f002:**
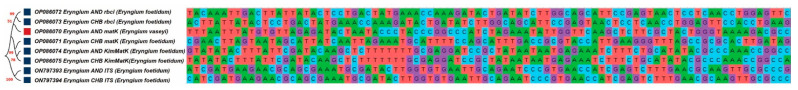
Multiple sequence alignment of *matK*, *Kim matK*, *rbcL*, and *ITS2* barcode primers and their phylogenetic relationship among two *Eryngium* genotypes.

**Figure 3 genes-13-01678-f003:**
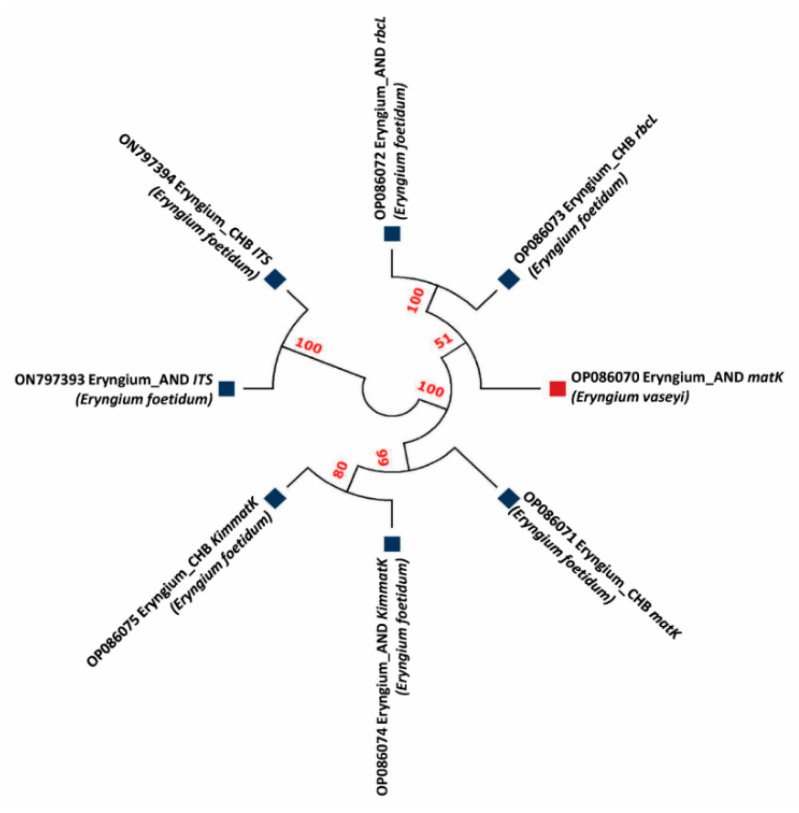
Maximum likelihood tree of *matK*, *Kim matK*, *rbcL*, and *ITS2* barcode primers depicting the phylogenetic relationship among two *Eryngium* genotypes. The bootstrap scores (1000 replicates) are shown (≥50%) for each branch.

**Figure 4 genes-13-01678-f004:**
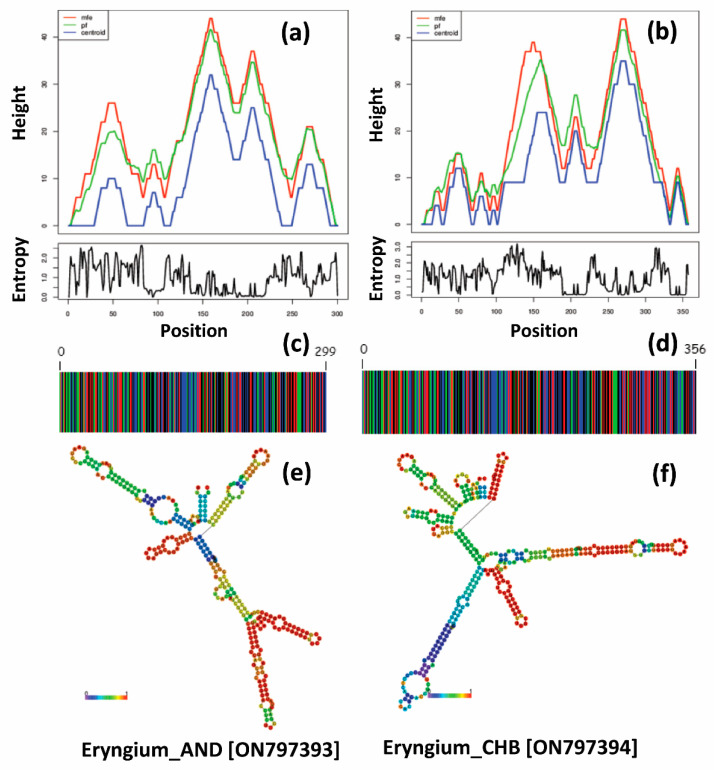
The variations observed in the predicted minimum free energy (MFE) (**a**,**b**), DNA barcodes (**c**,**d**), and secondary structures of *ITS2* region (consensus structure; **e**,**f**) for the *Eryngium* genotypes.

**Table 1 genes-13-01678-t001:** List of the barcoding primers used for PCR amplification of *Eryngium* spp.

Sl. No.	Region	Primer Name	Sequences (5′ to 3′)	Amplicon Size
1	*matK*	*matK XF*	TAATTTACGATCAATTCATTC	1500 bp
		*matK 5R*	GTTCTAGCACAAGAAAGTCG	
2	*Kim matK*	*3F_Kim matK*	CGTACAGTACTTTTGTGTTTACGAG	1500–1580 bp
		*1R_Kim matK*	ACCCAGTCCATCTGGAAATCTTGGTTC	
3	*rbcL*	*rbcLa F*	ATGTCACCACAAACAGAGACTAAAGC	600–800 bp
		*rbcLa R*	GTAAAATCAAGTCCACCRCG	
4	*ITS2*	*ITS2 S2F*	ATGCGATA CTTGGTGTGAATTATAGAAT	400–800 bp
		*ITS2 S3R*	GACGCTTCTCCAGACTACAAT	

**Table 2 genes-13-01678-t002:** Molecular identification of *Eryngium* sp. using *matK*, *Kim matK*, *rbcL*, and *ITS2* barcode genes.

Barcode Genes	Voucher Specimen	Scientific Name	Accession Number	E Value	Query Coverage	Percent Identity
*matK*	Eryngium_AND	*E. vaseyi*	OP086070	0.0	100%	99.76%
	Eryngium_CHB	*E. foetidum*	OP086071	0.0	100%	99.81%
*Kim matK*	Eryngium_AND	*E. foetidum*	OP086074	0.0	98%	99.74%
	Eryngium_CHB	*E. foetidum*	OP086075	0.0	100%	99.87%
*rbcL*	Eryngium_AND	*E. foetidum*	OP086072	0.0	100%	100%
	Eryngium_CHB	*E. foetidum*	OP086073	0.0	100%	100%
*ITS2*	Eryngium_AND	*E. foetidum*	ON797393	2 × 10^−153^	100%	100%
	Eryngium_CHB	*E. foetidum*	ON797394	2 × 10^−153^	100%	100%

**Table 3 genes-13-01678-t003:** Sequence characteristics, and the intraspecific and interspecific genetic divergence of candidate barcodes.

Voucher Specimen	Eryngium_AND				Eryngium_CHB			
Markers	*matK*	*Kim matK*	*rbcL*	*ITS2*	*matK*	*Kim matK*	*rbcL*	*ITS2*
Sequence length	823	763	493	300	534	780	476	357
Alignment length	823	763	492	300	533	773	419	357
Maximum score	1509	1399	909	555	979	1424	774	660
GC content (%)	37.18	36.70	43.00	62.33	35.96	36.28	43.28	63.31
CDS region	2	1	1	–	2	1	1	–
Amino acid	274	254	164	–	177	260	159	–

## Data Availability

Not applicable.

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
