# Peer review of "Molecular Phylogeny, DNA Barcoding, and ITS2 Secondary Structure Predictions in the Medicinally Important Eryngium Genotypes of East Coast Region of India"

_genes, 2022, doi:10.3390/genes13091678_

Round 1
Reviewer 1 Report
This article presented Molecular Phylogeny, DNA Barcoding, and ITS2 Secondary Structure Predictions in The Medicinally Important Eryngium Genotypes of East Coast Region of India. The study is well organized and data is well arranged. The findings would be helpful for future studies. There are some shortcomings for that should be resolve.
General comments
Abstract
Methods and main findings are not well presented in the abstract.
Introduction
The introduction part is well written but still some details are required. The authors should provide details of the taxonomic complexities and taxonomic position of the studied species.
Materials and methods
It would be much better to present picture of the studied plant in section 2.1
Section 2.4 PCR amplification could be cited with https://doi.org/10.2298/GENSR2002435A,
Discussion and conclusion
Discussion and conclusion are well presented. However, future recommendations based on the obtained results must be added in the conclusion section.
Author Response
Response to Reviewer 1 Comments
Reviewer’s comment: This article presented Molecular Phylogeny, DNA Barcoding, and ITS2 Secondary Structure Predictions in The Medicinally Important Eryngium Genotypes of East Coast Region of India. The study is well organized and data is well arranged. The findings would be helpful for future studies. There are some shortcomings for that should be resolve.
Authors’ response: Thank you very much for your kind remarks.
General comments
Abstract
Point 1: Methods and main findings are not well presented in the abstract.
Response 1: Thank you for your kind suggestion. Due to the word limit in the abstract section (200 words), we have precisely mentioned the method and findings. In methods, we have mentioned species discrimination using DNA sequencing with barcode genes (Lines 27-29). The DNA barcodes and ITS2 secondary structure has been predicted based on the minimum free energy (Line 32-33). The findings are presented in lines 29-35, along with the key recommendation (Line-33-35) and their implications (Line 35-37).
Introduction
Point 2: The introduction part is well written but still some details are required. The authors should provide details of the taxonomic complexities and taxonomic position of the studied species.
Response 2: Thank you for your kind suggestion. The taxonomic position and complexities have been updated in lines 42-45.
Materials and methods
Point 3: It would be much better to present picture of the studied plant in section 2.1
Response 3: Thank you for your kind suggestion. The picture of the studied species has been placed on the location map. As suggested, we have updated the pictures in Figure 1 (section 2.1).
Point 4: Section 2.4 PCR amplification could be cited with https://doi.org/10.2298/GENSR2002435A,
Response 4: Thank you for your kind suggestion. The reference has been cited in section 2.4 (Line 157-158) and updated in the reference section.
Discussion and conclusion
Point 5: Discussion and conclusion are well presented. However, future recommendations based on the obtained results must be added in the conclusion section.
Response 5: The future implications have been updated in lines 386-388.
We thank the expert reviewer for the constructive suggestions.
Reviewer 2 Report
The study is well presented however there are some shortcomings which must be addressed.
In abstract section the authors are mainly focused on background or justification. One to two sentences are enough, instead main findings should be displayed in abstract section.
In introduction also discuss about the origin of the studied genus.
Line 69-70 specify morphological characters and cite the respective studies.
The study is well designed but the sampling size is very poor. How, results and conclusion based on this sampling size can be justified. Add one or two reasons.
Line 130 present voucher numbers from the herbarium.
Line 272. Add citation the following study may be helpful. DOI: http://dx.doi.org/10.30848/PJB2022-3(19),
Discussion must be compared with recent studies on large scale.
Also add discussion about the ribosomal nuclear ITS2 region and its significance in phylogenetic.
Author Response
Response to Reviewer 2 Comments
The study is well presented however there are some shortcomings which must be addressed.
Authors’ response: Thank you very much for your kind remarks.
Point 1: In abstract section the authors are mainly focused on background or justification. One to two sentences are enough, instead main findings should be displayed in abstract section.
Response 1: Thank you very much for your suggestion. The background/rationale has been revised (lines 25-27), as suggested. Due to the word limit in the abstract section (200 words), we have precisely mentioned the background, rationale, methods, and findings. The findings are presented in lines 30-36, along with the key recommendation (Line-34-36) and their implications (Line 36-38).
Point 2: In introduction also discuss about the origin of the studied genus.
Response 2: Thank you very much for your suggestion. The centre of origin and its widespread has been mentioned in lines 51-55 (This perennial herb E. foetidum hailed from tropical America… Andaman and Nicobar Islands of India ).
Point 3: Line 69-70 specify morphological characters and cite the respective studies.
Response 3: Thank you very much for your suggestion. The morphological characters have been updated as suggested (Lines 71-75).
Point 4: The study is well designed but the sampling size is very poor. How, results and conclusion based on this sampling size can be justified. Add one or two reasons.
Response 4: The study focused on species discrimination in two important medicinally important Eryngium genotypes collected from the vulnerable east coast region of India (the Bay of Bengal and the Andaman Sea). Eryngium is a lesser-known crop gaining importance for medicinal, aromatic, and culinary purposes in this region. Proper identification using DNA barcoding techniques using chloroplast-plastid, and nuclear gene, in combination, would be a noble approach for correct species identification which would be useful for future breeding/crop improvement strategies in this potential crop. The standardization of DNA barcoding techniques, such as primer optimization with gradient PCR, PCR amplification, purification, and QC, has been performed with five replications to ensure reproducibility. The phylogenetic analysis was performed following the neighbor–joining tree and minimum evolution method with 1000 “Bootstrap phylogeny” test method and “kimura–2–parameter” substitution model (d–transitions) in MEGA software considering the transitional and transversional nucleotide substitution. ITS2 secondary structure predictions were performed after stringent sequence editing, barcode gap analysis, and multiple sequence alignment.
Point 5: Line 130 present voucher numbers from the herbarium.
Response 5: The herbarium voucher numbers obtained from the repository of ICAR-IIHR-CHES, Bhubaneswar, India are, 1. ‘Eryngium_CHB’, and 2. ‘Eryngium_AND’. However, the biological reference number, i.e., IC 0629514 has been obtained for ‘Eryngium_CHB’ from NBPGR, New Delhi; and the IC number for ‘Eryngium_AND’ has not been obtained as it is under consideration for release at CARI, Port Blair, IN.
Point 6: Line 272. Add citation the following study may be helpful. DOI: http://dx.doi.org/10.30848/PJB2022-3(19),
Response 6: Thank you for your suggestion. The reference has been cited in Line 279 and updated in the reference section.
Point 7: Discussion must be compared with recent studies on large scale.
Response 7: Thank you for your suggestion. Although, DNA barcoding studies in Eryngium are scanty, the discussion has been updated with recent studies on Lamiaceae as suggested (Lines 316-320 and Lines 336-337).
Point 8: Also add discussion about the ribosomal nuclear ITS2 region and its significance in phylogenetic.
Response 8: Thank you for your suggestion. Discussion on the significance of ITS2 in the phylogenetics relationship has been updated (Lines 338-342).
We thank the esteemed reviewer for the valuable comments/suggestions.